# Cooling Effect of Trees with Different Attributes and Layouts on the Surface Heat Island of Urban Street Canyons in Summer

Shaojun Yan [1], Tailong Zhang [1], Yu Wu [1], Chu Lv [1], Feng Qi [1], Yangen Chen [2], Xiaohua Wu [1] and Yamei Shen [1,*]

[1] School of Landscape Architecture, Zhejiang Agricultural and Forestry University, Hangzhou 311300, China
[2] Lin'an District Agricultural and Rural Bureau, Hangzhou 311300, China
* Correspondence: yameishen@zafu.edu.cn

**Abstract:** In recent years, the impact of surface heat islands in urban street canyons has become increasingly apparent. However, the research on the use of trees to mitigate surface heat islands remains limited. To address this gap, this study combines experiments and simulations to analyze the cooling effect of trees on surface temperatures under varying timeframes and layouts in an east–west street canyon. The results reveal that the temperature of the road decreases by 10–15 °C, which is 2–4 times greater than that on the south side. Moreover, at 5:00 p.m. in the afternoon, the cooling effect on the south side is 10.3 °C, which is twice that of the north side. In practical planning and design, the diameter of the tree canopy should be maximized, and trees with leaf-area densities greater than 1.5 $m^2/m^3$ should be selected. Additionally, the layout of trees should be optimized to maximize the tree canopy coverage. These findings have important implications for optimizing plant selection and placement in street canyons.

**Keywords:** street canyon microclimate; green space; ENVI-met; surface temperature; tree cooling

## 1. Introduction

As urban areas continue to grow in population density and development, the prevalence of impervious surfaces is increasing and exacerbating the heat island effect [1]. This phenomenon is having significant impacts on urban microclimates and posing a threat to the livability and well-being of urban inhabitants [2,3]. The intensity of surface heat islands is more pronounced in areas with high density mid-rise and low-rise buildings [4]. In particular, the intensity of the surface urban heat island (SUHI) is higher during daytime hours in most cities, exacerbating building cooling loads and potentially contributing to increased mortality rates [5–7]. Moreover, the surface heat parameters vary for different materials, especially for asphalt pavements which possess a dark color that absorbs significant amounts of solar radiation and exhibits viscoelastic behavior, making them an important source of heat generation in urban canyons [8]. The rheological characteristics of asphalt surfaces under high temperature conditions and their high viscosity also contribute to the formation of vehicle ruts [9]. Thus, mitigating the urban surface heat island during daytime hours represents a pressing challenge that requires immediate attention from the scientific and academic communities.

The urban surface heat island is influenced by various factors such as urban geometry, building density [10], land cover type [11] and urban functions [12]. Several strategies have been proposed to mitigate the heat island effect. One approach is to increase the aspect ratio of building height to street width, which increases the shaded area and reduces the surface heat island [13]. Increasing building height and providing additional shading also decreases the surface sky view factor (SVF), which reduces the absorption of solar radiation and ultimately leads to a decrease in surface temperature [14,15]. Increasing surface albedo has been shown to have a significant cooling effect [16,17] and can also reduce mortality rates [18]. In addition, trees are also effective in reducing ambient temperatures through

their shading effect [19], making the optimization of green spaces and the strategic planting of trees a key strategy in mitigating the urban surface heat island [20].

The impact of tree shading on solar radiation received at the ground comprises both direct shortwave solar radiation and sky scattering [21]. The shading effect of tree canopies has been analyzed through the modeling of urban canopies [22,23], where the condition of the foliage is a critical aspect in the exchange of radiation and cooling [24]. Different foliage conditions can impact the magnitude of transmittance, with trees having a low transmittance able to shade more shortwave radiation and reduce ground-absorbed energy for cooling purposes [25]. Trees with high leaf-area density (LAD) and leaf-area index (LAI) tend to have a lower transmittance and provide more effective shading of solar radiation [26]. The cooling effect of trees with varying LAD was analyzed by Tsoka et al. who find that the cooling capacity of trees was closely related to LAD [27]. Fahmy et al. observed that in low and mid latitude areas, trees with LAI less than 1 have a solar radiation interception rate of less than 0.5 [28]. Based on this characteristic, the use of ENVI-met software has become a common approach to simulating the impact of tree characteristics on a microclimate [29–32]. By incorporating different LAD and LAI of trees into the simulations, the thermal environment of urban streets can be analyzed, including the cooling benefits of street trees [33], thermal comfort [34–36] and air temperature cooling [37]. Such simulations are important for the regulation of street microclimates and the mitigation of urban heat islands. Furthermore, the spatial arrangement of trees can significantly impact the surrounding thermal environment.

By studying the thermal radiation disturbance of trees on the wall at different locations on the west side of the building, Zhang et al. find that the cooling effect of trees facing the wall with larger angular coefficients is better [16]. In a similar vein, F. Calcerano et al. utilize a genetic algorithm to determine the cooling effects of trees oriented in various directions and have made practical recommendations for their implementation [38]. The green coverage of trees is an important factor that influences surface cooling, and as the green coverage of trees in streets continues to increase, street temperatures decrease nonlinearly [39]. He et al. [40] point out that when considering plant green coverage, spatial and temporal characteristics should also be taken into account to maximize its cooling effect.

However, the preceding studies have primarily focused on the reduction of temperature by trees and energy conservation in buildings, while neglecting the crucial aspect of how trees cool the surrounding ground, which is an important link in the study of tree mitigation of surface heat islands. Furthermore, there is a lack of trees to explore separately for the cooling of asphalt motorways as well as paved roads.

In view of this, this study employs a combination of simulation and field measurements to investigate the cooling characteristics of trees on surface heat islands, as well as to compare the cooling benefits of trees with different attributes and spatial layouts. The cooling of trees on asphalt motorways as well as on paved roads is also discussed and studied separately. The findings of this study serve as a scientific reference for selecting tree species for street greening and green space planning and design.

## 2. Methods

Field experiments can be used to study the cooling effect of plants under different scenarios, but the experimental period is long and it is not easy to control variables. Therefore, this study uses a combination of experiments and simulations to analyze and study the impact of different tree planting scenarios on the surface heat island. This study is divided into two steps. The first step is to select typical street representatives of cities in the hot summer and cold winter zone and obtain the meteorological and surface temperature data of the experimental site under clear and cloudless weather. At the same time, the same simulation model as the actual scene is established in the simulation software ENVI-met 4.4 version, and the meteorological data in the experiment is used for simulation. ENVI-met is a 3D grid simulation software based on fluid dynamics and thermodynamics, which

was developed by Prof. Bruse and his colleagues in Germany and is widely used in the field of urban microclimate simulation [41]. After the simulation is over, the RMSE of the simulated and measured ground temperature is calculated, and the simulation method is continuously optimized to the best effect according to the calculation results. In the second step, using the final optimized simulation method, different vegetation attributes and spatial layout simulation scenarios were established to explore the pattern of surface temperature changes in different scenarios. According to the simulation results, the street vegetation tree species selection and layout optimization scheme is proposed (Figure 1).

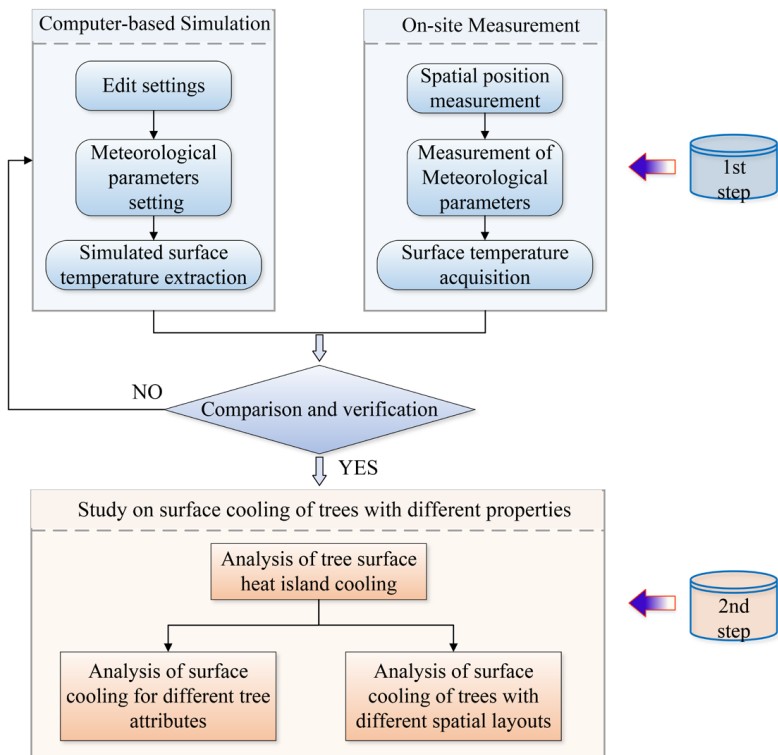

**Figure 1.** Flow chart of the simulation method.

### 2.1. On-Site Measurement

Hangzhou, located in the northwest of Zhejiang Province, China, is bounded by latitude 29°11′~30°34′ N and longitude 118°20′~120°37′ E. The climate of Hangzhou belongs to the subtropical monsoon climate, with four distinct seasons, mild and humid, sufficient sunshine, and abundant rainfall, including the characteristics of both subtropical and northern subtropical [28]. In the context of global warming, the average summer temperature in Hangzhou has been in a linear upward trend since the 1970s. According to the statistics of high temperature in Zhejiang province from 1951 to 2022, the high temperature in Hangzhou occurred mostly from June to September, with the highest temperature reaching 43.2 °C [42]. From 2000 to 2020, the high impact area of surface heat island in Hangzhou increased at the edge of the city [43,44].

In this study, the greening space characteristics of street canyons were taken as the research object, and the influence of variables such as the trend of street canyons, the ratio of height to width, and the form of cross section, etc., should be excluded. Therefore, Fengqi Road was selected as a representative street canyon space (Figure 2). The street canyon is located in the bustling area of Shangcheng District, Hangzhou, with a large pedestrian flow. The Fengqi Road, with a total length of 2.86 km and a street width of 40 m, consists of paved roads that are, respectively, 8 m wide and asphalt lanes that are 24 m wide. For our experiment, we selected a main measurement area of 130 m in length from the aforementioned street. Moreover, the sides of the street canyons are residential

buildings with 7–8 floors in height, and the ratio of height to width of the street canyons is almost the same (Figure 3).

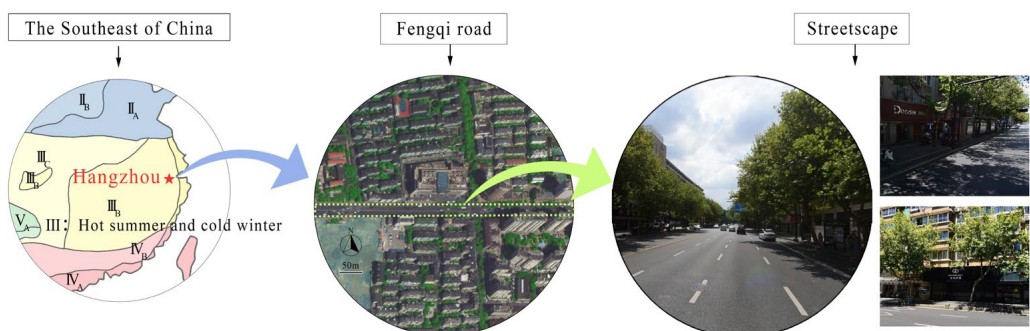

**Figure 2.** Location of street canyon.

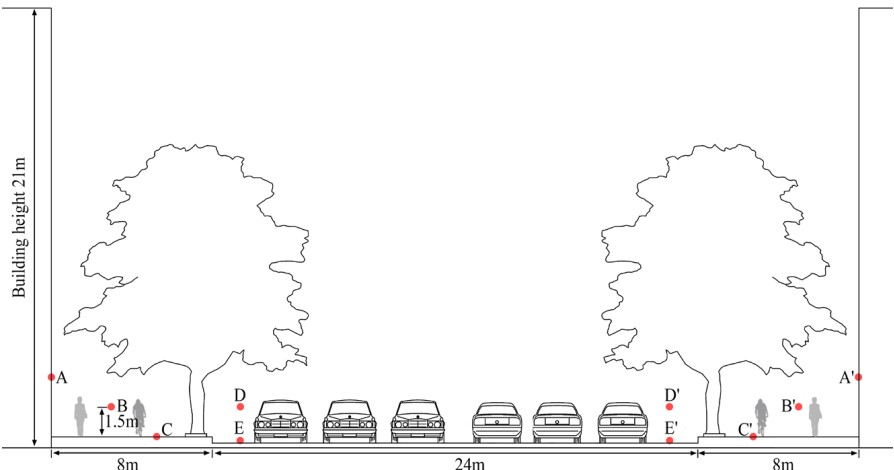

**Figure 3.** Distribution map of each measuring point on Fengqi road.

The data collection was carried out in the summer of 2019, specifically in July, August and September. During the months of July, August and September, Hangzhou City experiences an average precipitation of 207.9 mm, 246.5 mm, and 105.4 mm, respectively. The mean monthly temperatures during these months are 28.1 °C, 29.4 °C and 25.1 °C, while the mean monthly sunshine hours are 162.5 h, 228.5 h and 189.7 h. During this period, the city is frequently influenced by subtropical high-pressure systems, resulting in a quasi-stationary wind environment under clear skies [42]. The study focuses on the microclimate of street valley areas and aims to exclude the impact of variables such as wind direction and average wind speed. The experimental dates were July 23rd and 24th, August 16th and 17th, and September 9th and 10th. Three days with less cloud interference were selected as the focus of this study for analysis. To ensure the accuracy of the data, repeated tests were conducted on three sections of the north and south sides of each street valley, with measurements taken at five different times a day, at 2-h intervals, from 9:00 a.m., 11:00 a.m., 1:00 p.m., 3:00 p.m. and 5:00 p.m. (Figure 3). The microclimate data collected included building surface temperature (point A and A') of the street valley, air temperature and humidity at 1.5 m in both the pavement (point B and B') and motorway (point D and D'), surface temperature of the pavement (point C and C') and motorway (point E and E'), black globe temperature, wind direction, and average wind speed. The study employed TES1361C equipment to measure air temperature and humidity, TES-1333 equipment to measure solar radiation and FLUKE F59 equipment to measure surface temperatures of both the ground and building exteriors. For each type of measurement within each section, three sampling points were selected to avoid significant data deviations. A control group with no green coverage was also measured in each valley.

### 2.2. Computer-Based Simulation

In this study, ENVI-met was employed for the purpose of numerical simulation verification. A three-dimensional model of the street canyon scene was established by inputting the basic information, including site size, geographical coordinates and orientation, as derived from the results of a preliminary survey (Figure 4). Based on previous case studies, it was determined that a meteorological condition with limited cloud coverage and gentle breezes in fine weather would be appropriate for the simulation. Therefore, three dates in the summer of 2019 (23 July, 16 August and 10 September) were chosen for the simulation and initial conditions were established based on measured weather data (Table 1). The geographic coordinates were set at 119.72° E and 30.23° N, with a time zone of UTC/GMT+08:00. The simulation model consists of a total of 100 × 80 × 30 grids, with dimensions of 2 m × 2 m × 2 m for each grid. To enhance the accuracy of radiative heat transfer calculations, we employed the Indexed View Sphere (IVS) module in our simulation setup. The simulation results were compared to the measured surface temperature data to evaluate the reliability of the simulation software. If the simulation is found to be reasonable and reliable, additional scenarios will be simulated in the flowing.

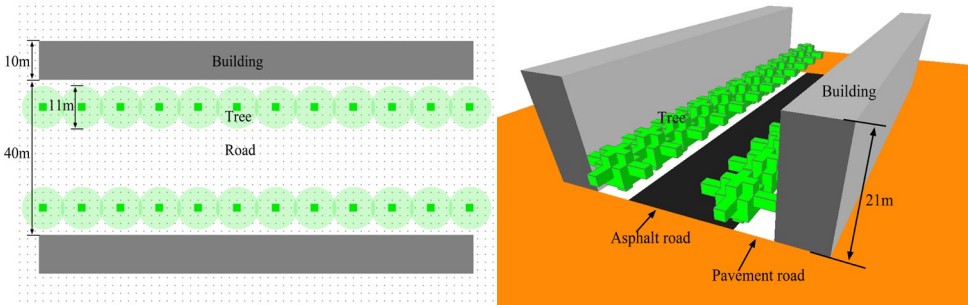

**Figure 4.** Three-dimensional model of a typical street-valley plot sample.

**Table 1.** Climate conditions for measurement and simulation.

| | Simulation Parameters Conditions | | |
| --- | --- | --- | --- |
| | **23 July** | **16 August** | **10 September** |
| Max air temperature(°C) | 39.2 | 38.8 | 34 |
| Min air temperature(°C) | 29 | 29.4 | 22 |
| Max relative humidity (%) | 77 | 69 | 43 |
| Min relative humidity (%) | 35 | 39 | 76 |
| Foliage Short-wave Albedo | 0.3 | 0.3 | 0.3 |
| Short-wave radiation absorption | 0.5 | 0.5 | 0.5 |
| Duration (h) | 24 | 24 | 24 |
| Start time (Local) | 07:00 | 07:00 | 07:00 |

### 3. Results

#### 3.1. Field Observation Verification

The comparison of the measured and simulated values of surface temperature reveals that the root mean square error (RMSE) between the two is within an acceptable range (Figure 5). Specifically, the RMSE values for the hotter months of July and August are smaller, indicating a better simulation [27,45,46].

The analysis of the errors in the data suggests that both objective and subjective factors may have contributed to the results. Objective factors such as cloud variation and wind speed variation due to traffic flow may have influenced the accuracy of the measurements. Subjective factors, such as the walking speed of the individual using the handheld instruments, may have also had an effect on the measured values of temperature, humidity and wind speed. Furthermore, certain limitations of the simulation software may have also played a role.

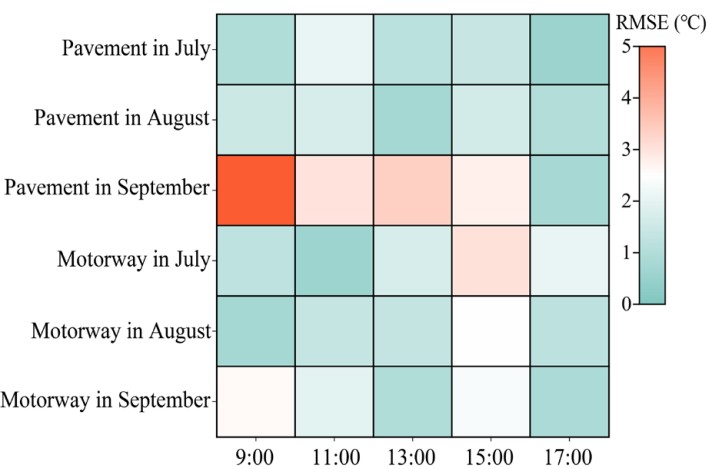

**Figure 5.** Verification of simulation and measurement.

Despite these errors, the simulation results are deemed reasonable and reliable. As such, the simulation software may be utilized to further examine the cooling mechanism of trees in street canyons and the cooling effect of trees with different attributes.

*3.2. Numerical Simulation Results*

3.2.1. Surface Cooling of Trees at Different Time Points

The limited number of measurement points in the experiment precludes a comprehensive representation of the overall ground temperature distribution in the street. Therefore, using the same simulation results as the real scenario, the cooling mechanism of the tree was analyzed. The simulation results of the average ground temperature of a road on 23 July were extracted, taking into account the shadows cast by buildings at various times. This study finds that the cooling effect of trees varies depending on the time of day (Figure 6). For instance, at 9 a.m., the buildings cast fewer shadows in the street canyon, resulting in a similar ground temperature and cooling effect on both sides of the street. The average ground temperature of the paved road without trees is approximately 31 °C, while the motorway temperature is about 33 °C. However, with the presence of trees, the average ground temperature drops to around 27 °C, and the motorway temperature decreases to about 30.5 °C.

Interestingly, the average surface temperature of the roadway at each location increases and then decreases between 11:00 a.m. and 3:00 p.m., with the cooling effect of trees being more prominent on the north side of the street. During this time, buildings on the south side cast some shadows on the pavement, resulting in a cooling effect of trees on the north side that reached 10–15 °C, which is about 2–4 times more effective than the cooling effect on the south side. With the exception of the pavement on the south side, the other three roads are less affected by building shadows. In the absence of tree planting, the average surface temperature of the three roads is similar. However, after trees are planted, the cooling effect of the paved road is higher than that of the motorway, reaching 10–15 °C, while the cooling effect on both sides of the motorway is between 5.3 °C and 6.1 °C.

At 5:00 p.m., when buildings on the north side produce shadows in the street canyon, the surface temperature of the paved road on the south side is 2.5 °C higher than that on the north side. The trees re able to cool the paved road on the south side by 10.3 °C, about two times more effective than the paved road on the north side. The cooling effect on the motorway is similar, ranging from 4.5–5.2 °C.

When planning and designing street canyons, it is important to consider the shading conditions of the buildings on both sides of the street. By calculating the shading conditions of the buildings at different times of the day, the planting distance between the trees and buildings can be optimized for maximum cooling effect. This approach can effectively reduce the urban heat island effect and improve thermal comfort in the urban environment.

Therefore, it is crucial to incorporate these considerations into street canyon design for sustainable, comfortable and livable cities.

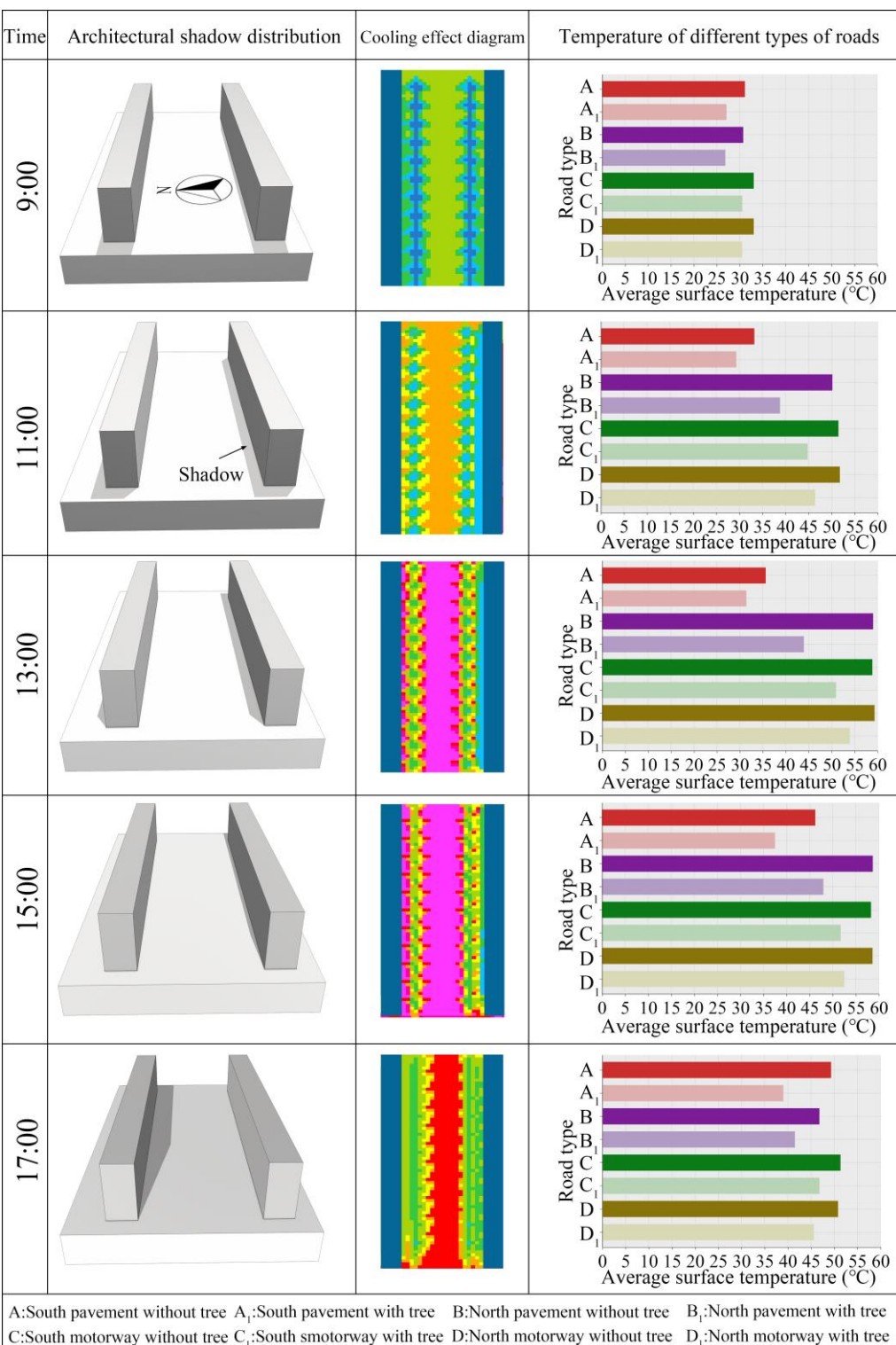

**Figure 6.** Analysis of the cooling effect at each time period.

The cooling values of trees in each location along the street were analyzed to generate Figure 7, which displays the thresholds and trends of tree cooling during each time period. The figure shows that the majority of the cooling effects of trees fall within the 0–2 °C range for each time interval. It is noteworthy that there are fewer data points without cooling,

indicating that even in areas where trees do not provide shade, the transpiration of trees and their shading effect on shortwave diffuse radiation from the sky and shortwave and longwave radiation from building surfaces also have a certain degree of a cooling effect.

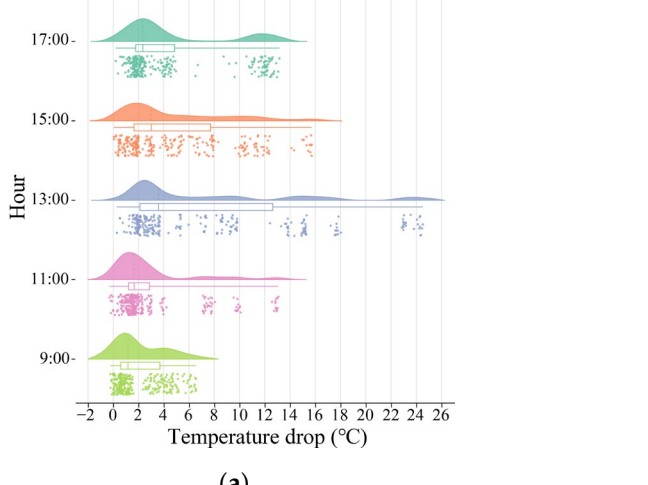 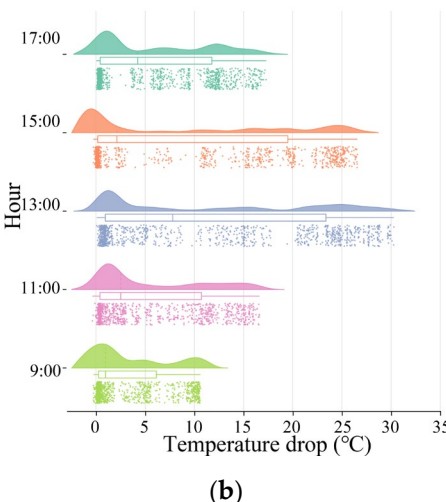

**(a)**                                                 **(b)**

**Figure 7.** Distribution of cooling effect at each location of street canyon, (**a**) pavement and (**b**) motorway.

The minimum and maximum thresholds of the cooling effect of trees in the street initially increases and then decreases over time, reaching their peak at 1:00 p.m. Correspondingly, the ground temperature is also higher during this time, resulting in a high threshold of cooling points (i.e., cooling effect >20 °C) in this location.

Overall, the cooling patterns of trees on motorways and paved roads are similar, with the cooling effect of trees on motorways being slightly higher than that of paved roads. Although trees effectively alleviate urban surface heat islands, their positive effect on pedestrians and vehicles is relatively limited in low-cooling areas. Therefore, further research is needed to explore how to increase the high-cooling area efficiently, taking into account different attributes and layouts of trees for surface cooling.

### 3.2.2. Surface Cooling Effect of Different Tree Attributes

To further explore the cooling of surface heat island by different tree attributes, the same model scenarios with tree crown diameters ($D_C$) of 7 m, 9 m, 11 m and LAD of 0.5 $m^2/m^3$, 1 $m^2/m^3$, 1.5 $m^2/m^3$, 2 $m^2/m^3$ and 2.5 $m^2/m^3$ were simulated using meteorological conditions and street models on July 23, and the surface cooling values were extracted for each simulation result (Figure 8). Figure 8a shows the increase in vertical projection of trees as a quadratic function of the increase in crown size when the tree canopies do not overlap. The green cover of the trees increases from 21% to 34% and 45%, respectively, and their cooling effect is then significantly enhanced. And when the crown diameter of trees reaches 11 m, the crown diameter of adjacent trees begin to overlap due to the distance between trees of 10 m. At this time, the tree crown diameter continues to increase, the growth rate of greenery coverage will slow down, and the cooling effect will be enhanced at a relatively reduced rate. On the other hand, the cooling effect of trees gradually increases with the increase of LAD. When the LAD reaches 1.5 $m^2/m^3$, the peak cooling effect exceeds half of the peak difference between LAD of 2.5 $m^2/m^3$ and LAD of 0.5 $m^2/m^3$ (indicated by the red dashed line), and the rate of change in the cooling effect of trees begins to slow down (Figure 8b). Although the cooling effect also increases with the further increase of LAD, the magnitude of the increase becomes smaller. Therefore, to alleviate the urban heat island effect and improve pedestrian thermal comfort, it is recommended to select trees with a LAD value of 1.5 $m^2/m^3$ or greater in planning and design.

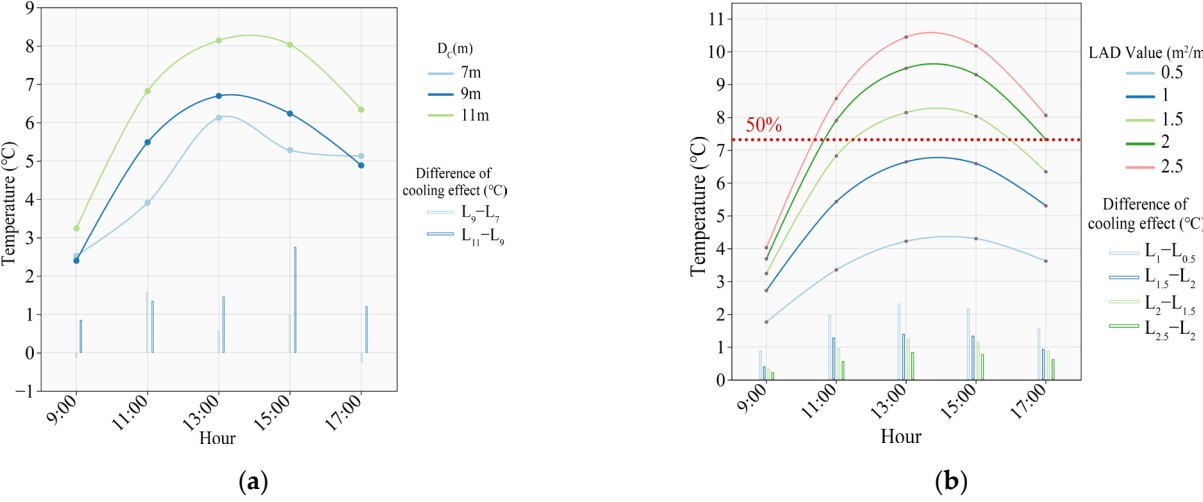

**Figure 8.** Cooling effect of different tree attributes (**a**) $D_C$ and (**b**) LAD.

### 3.2.3. Surface Cooling of Different Spatial Changes in Trees

To investigate the mitigation of surface heat islands using different tree layouts, three groups were modeled using ENVI-met (Figure 9). All groups of models used trees with a crown diameter of 11 m, height of 11 m, and LAD of 1.5 $m^2/m^3$. In the first group, the cooling effect of trees was explored by changing the tree quantity to alter the greening coverage. From S1 to S4, the distance between trees decreased by 2 m in each scenario, resulting in an increase in the number of trees. The second group consisted of 12 trees with various configurations, including single standalone trees, two trees planted together, three trees planted together and four trees planted together. This group aims to explore the cooling effect of different combinations of trees while ensuring the total number of trees remained constant. Finally, the third group used the same tree layout and attributes as scenario 5 but varied the building height to investigate the impact of different building heights on surface temperature.

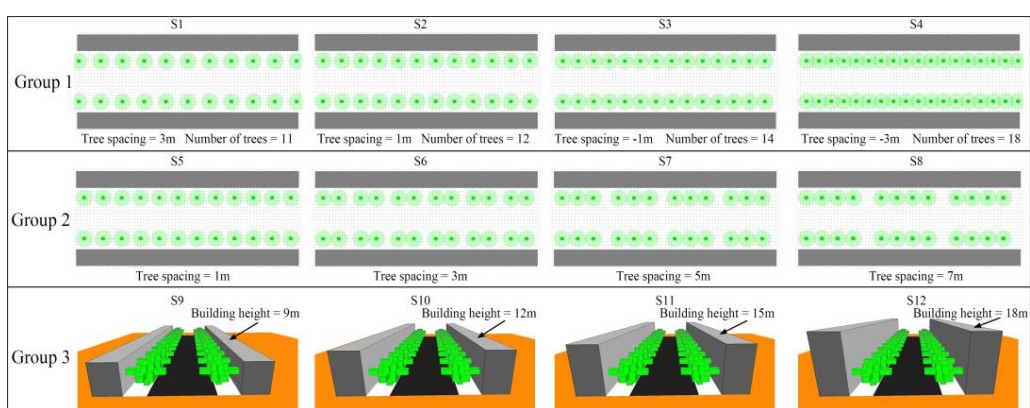

**Figure 9.** Simulation scenarios for different tree layouts.

According to the first simulation results, the cooling effect of trees increases significantly with an increase in the number of trees and a decrease in canopy spacing, across all time periods (Figure 10). The cooling clouds diagrams generated by trees show that areas with higher cooling effects increase notably, particularly between 11:00 a.m. to 3:00 p.m. The areas with high cooling effects at each location change from early to late as the shadow of the trees changes, gradually moving from west to east. The change in green coverage is positively correlated with the change in tree cooling effect, with correlation coefficients of 0.945, 0.961, 0.978, 0.965 and 0.955 at 9 a.m., 11 a.m., 1 p.m., 3 p.m. and 5 p.m., respectively. This indicates that the green cover of trees is a crucial factor that influences the cooling

effect of trees. In practical planning, adjusting the layout and canopy size of trees can enhance their cooling effect by maximizing the green cover of the tree canopy.

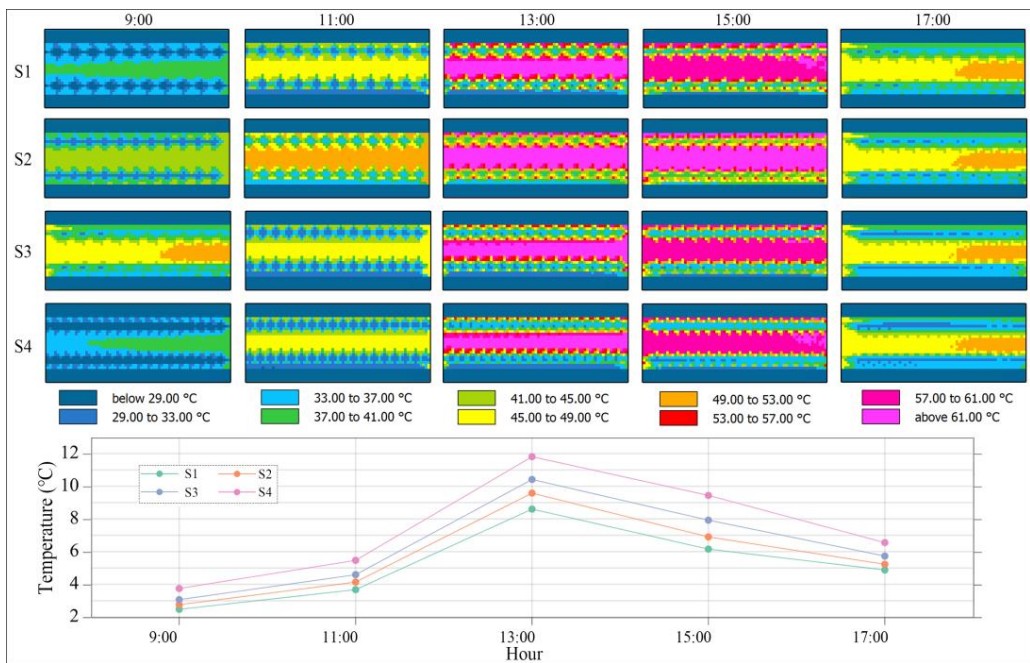

**Figure 10.** Surface Cooling Effects of Different Tree Crown Spacing.

In the second and third groups of simulations, the placement of trees and height of buildings were adjusted, respectively, yet neither modification impacted the green coverage of trees on the street. Figures 11 and 12 illustrate that both groups yield comparable ground cooling effects despite variations in simulated conditions, and no significant changes are observed. This reaffirms the critical role of tree coverage in mitigating ground temperatures.

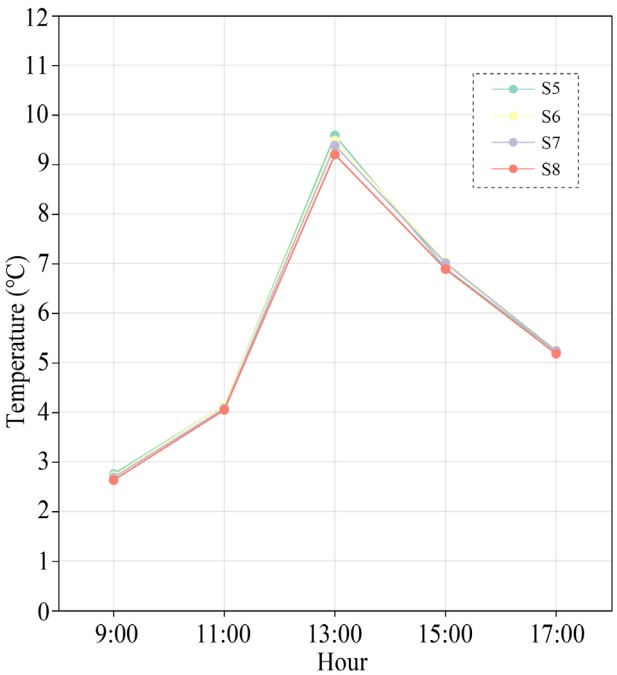

**Figure 11.** Surface cooling effects at different tree locations.

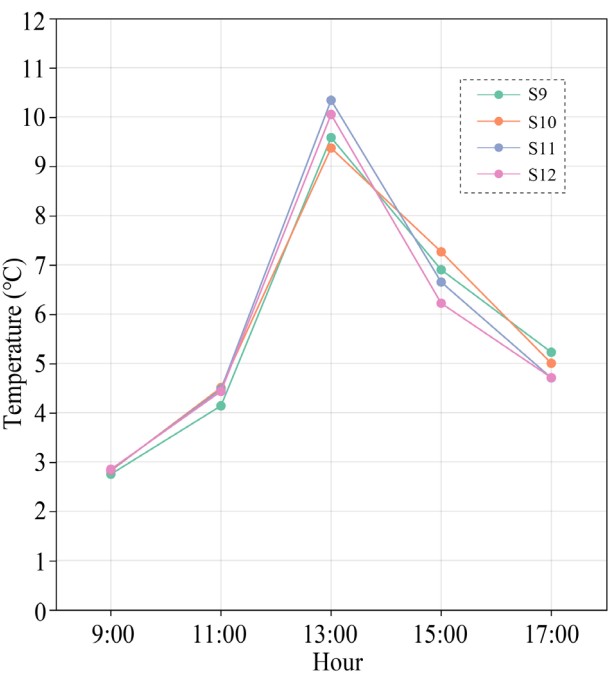

**Figure 12.** Surface cooling effects at different building heights.

## 4. Discussion

Currently, research on surface heat islands is conducted at various scales. Large-scale studies often use remote sensing images to infer surface temperature and analyze the urban surface heat island effect. For example, Landsat satellite images have been used to reveal that green surfaces can provide cooling effects ranging from 5–10 °C [6,44,47]. While these studies provide valuable information for decision-makers at large scales, ground temperature inversion using remote sensing imagery still lacks precision in street-valley scales. Other studies had focused on analyzing the contributions of individual trees to the environment [21], but urban greening is a collective effort that reflects the contributions of a community; obtaining accurate data that truly represents human perceptions is challenging. This study focuses on the contribution of street-level greenery to environmental temperature, taking into account human height and environmental factors. Specifically, we investigate the cooling effects of tree attributes and spatial layout on surface temperature, providing more robust evidence for urban greenery construction from the perspectives of humans and their environment.

Simulation results indicate that trees play a crucial role in cooling the urban canyon's surface. The best cooling effect occurs between 1 p.m. and 3 p.m., consistent with Duval et al.'s findings [48]. It is noteworthy that the cooling effect on asphalt motorway can reach up to 30.28 °C, while it can reach up to 24.5 °C on laid pedestrian roads. Moreover, Sharmin et al. find that trees are capable of cooling black surfaces down to 28.1 °C [49]. This difference may be due to the fact that the summer maximum temperature in their studied city ranged from 28–30 °C, whereas in Hangzhou, where this study was conducted, the summer maximum temperature can exceed 40 °C. During our experiment, the highest temperature reached 39 °C, demonstrating that trees provide better cooling effects on high-temperature asphalt surfaces. Tun et al. discover that shading reduces surface temperatures by an average of 20 °C, with the maximum temperature decreasing by 40 °C. Their study site had higher temperatures than ours, with the highest temperature in New South Wales, Australia, reaching 48.9 °C in January 2020 [50]. Therefore, the cooling effect of trees is largely influenced by urban heatwaves.

Furthermore, different tree attributes have varying mitigating effects on the urban surface heat island. In this study, we find that trees with larger crown have better cooling effects, consistent with Wang et al.'s results [51]. This is because, as the canopy size

increases, the sky view factor below the tree decreases, reducing the amount of solar radiation received. Additionally, LAI and LAD are crucial factors. Hardin and Jensen's research shows that an increase in LAI of one unit could lower surface temperatures by 1.2 °C [52], while a 1 $m^2/m^3$ increase in LAD reduces surface temperatures by 4.63 °C. In our study, an increase of 0.5 units in LAD lowers surface temperatures by 0.9–2.2 °C, which is similar to the previous findings. However, we find that the cooling effect of trees decreases as LAD increases due to the nonlinear reduction in transmittance. When LAD reaches a certain size, trees become denser, and increasing LAD has little impact on their light transmission, resulting in a less significant increase in shading.

In addition to the inherent properties of trees, the diverse layout of trees can also have different impacts on surface temperature. Research has found that as the number of trees increases, the tree crown spacing decreases and green coverage increases. The cooling effect of the trees is best when their crowns overlap. The study results are consistent with Narimani's finding that overlapping trees have a better cooling effect [53]. At the same time, as the crowns of the trees overlap, the rate at which trees cool the environment gradually decreases. This is because the crowns of other trees weaken the shading potential of adjacent trees. Additionally, building height also has an important impact on the cooling effect of trees. Previous research has shown that buildings have a certain inhibitory effect on the shading of trees [53]. This study simulates the effects of different building heights on the cooling effect of trees, and finds that the impact is relatively small. This may be due to the high solar altitude angle and the east–west orientation of the experimental streets in this study, resulting in a relatively small impact of building shadows. Further research will explore the impact of building height on tree cooling for streets oriented north–south and at different dates and times.

## 5. Conclusions

A combined experimental and simulation study was conducted to investigate the cooling effect of trees on the street canyon surface heat island under various conditions. The following conclusions can be drawn from the results:

(1) The reliability of the ENVI-met software in simulating street canyon surface temperatures is validated through comparison with measured data.

(2) The analysis of the cooling effect of trees on the north and south lanes of the street canyon at different times indicates that on the asphalt of the motor vehicle lane, the cooling effect can reach up to 30.28 °C, while on the paved road, it can reach up to 24.5 °C. In the daytime, the cooling effect is consistent at 9 a.m. However, between 11 a.m. and 3 p.m., the cooling effect of trees on the northern side of the road surface ranges from 10 °C to 15 °C, which is 2 to 4 times that of the southern side. At 5 p.m., the cooling effect on the southern side reaches 10.3 °C, which is twice that of the northern side.

(3) Through an analysis of tree parameters and spatial layout variations, it has been determined that maximizing the canopy diameter and selecting trees with an LAD greater than 1.5 $m^2/m^3$ can aid in reducing the surface heat island effect in urban street canyons as part of urban planning and design. Additionally, adjusting the layout while keeping the number of trees constant has a relatively minor impact on the average ground temperature of the road. However, increasing the number of trees and thus improving the green coverage of trees significantly enhances the cooling effect.

**Author Contributions:** Conceptualization, S.Y. and T.Z.; methodology, S.Y. and X.W.; software, T.Z. and Y.S.; validation, C.L., F.Q. and S.Y.; formal analysis, X.W.; investigation, S.Y.; resources, Y.S.; data curation, T.Z. and C.L.; writing—original draft preparation, S.Y.; writing—review and editing, F.Q.; visualization, Y.W.; supervision, Y.C.; project administration, X.W.; funding acquisition, Y.S. All authors have read and agreed to the published version of the manuscript.

**Funding:** This research was funded by the National Natural Science Foundation of China (Grant No.5197081040).

**Institutional Review Board Statement:** Not applicable.

**Informed Consent Statement:** Not applicable.

**Data Availability Statement:** The data presented in this study are openly available in FigShare at [https://figshare.com/articles/dataset/dx_doi_org_10_6084_m9_figshare_6025748/6025748] (accessed on 5 May 2023).

**Conflicts of Interest:** The authors declare no conflict of interest.

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
