# Peer review of "Cooling Effect of Trees with Different Attributes and Layouts on the Surface Heat Island of Urban Street Canyons in Summer"

_atmosphere, doi:10.3390/atmos14050857_

Round 1
Reviewer 1 Report
This study analyzes the impact of different tree planting scenarios on the urban surface temperature using a combination of experiments and simulations. Based on the simulation results, a reference for tree species selection and layout optimization scheme is proposed. This manuscript has a detailed explanation of the results, but it still needs further improvements. Here are some comments that I would recommend:
1. Existing studies have simulated the cooling effect of tree layout, leaf area densities, and green coverage. The authors need to check the literature and clearly state what this paper builds upon earlier findings and achievements. I also suggest adding more information by comparing the results of this study to the related studies in the discussion section.
2. Please provide a better explanation for why the surrounding ground is more important. In large-scale urban environments, thermal comfort has already taken the influence of the surrounding ground into account.
3. Section 3 to Section 5? Incorrect section numbering. (P148-P281)
This study analyzes the impact of different tree planting scenarios on the urban surface temperature using a combination of experiments and simulations. Based on the simulation results, a reference for tree species selection and layout optimization scheme is proposed. This manuscript has a detailed explanation of the results, but it still needs further improvements. Here are some comments that I would recommend:
1. Existing studies have simulated the cooling effect of tree layout, leaf area densities, and green coverage. The authors need to check the literature and clearly state what this paper builds upon earlier findings and achievements. I also suggest adding more information by comparing the results of this study to the related studies in the discussion section.
2. Please provide a better explanation for why the surrounding ground is more important. In large-scale urban environments, thermal comfort has already taken the influence of the surrounding ground into account.
3. Section 3 to Section 5? Incorrect section numbering. (P148-P281)
Author Response
Dear reviewer,
We sincerely appreciate the editor’s responsible work and all reviewers' thoughtful comments and suggestions. Your ideas have enabled us to improve the quality of research work and make our article more scientific and clear. We have carefully reviewed the comments and have revised the manuscript accordingly. Our responses are given in a point-by-point manner below. The manuscript was modified using track Changes. Finally, we would like to take this opportunity to wish all the journal editors and reviewers a wonderful season and a happy 2023!
Reviewer 1:This study analyzes the impact of different tree planting scenarios on the urban surface temperature using a combination of experiments and simulations. Based on the simulation results, a reference for tree species selection and layout optimization scheme is proposed. This manuscript has a detailed explanation of the results, but it still needs further improvements. Here are some comments that I would recommend:
- (1) Existing studies have simulated the cooling effect of tree layout, leaf area densities, and green coverage. The authors need to check the literature and clearly state what this paper builds upon earlier findings and achievements. I also suggest adding more information by comparing the results of this study to the related studies in the discussion section.
- Many thanks for yoursuggestions. Early discoveries and related research have added references in the introduction, and a separate discussion section has been added to the manuscript. (line 29; line 33; line 77; line 85; line 90; line 322;).The newly added discussion is as follows:
Currently, research on surface heat islands is conducted at various scales. Large-scale studies often use remote sensing images to infer surface temperature and analyze the urban surface heat island effect. For example, Landsat satellite images had been used to reveal that green surfaces can provide cooling effects ranging from 5-10℃ [6,43,46]. While these studies provide valuable information for decision-makers at large scales, ground temperature inversion using remote sensing imagery still lacks precision in street-valley scales. Other studies had focused on analyzing the contributions of individual trees to the environment [21], but urban greening is a collective effort that reflects the contributions of a community; obtaining accurate data that truly represents human perceptions is challenging. This study focuses on the contribution of street-level greenery to environmental temperature, taking into account human height and environmental factors. Specifically, we investigated the cooling effects of tree attributes and spatial layout on surface temperature, providing more robust evidence for urban greenery construction from the perspectives of humans and their environment.
Simulation results indicated that trees play a crucial role in cooling the urban canyon's surface. The best cooling effect occurs between 1 p.m. and 3 p.m., consistent with Duval et al.'s findings [47]. Notably, on motorway asphalt pavements, the cooling effect reaches up to 30℃, while Sharmin et al. found that trees could cool black surfaces up to 28.1℃ [48]. This difference may be due to the fact that the summer maximum temperature in their studied city ranged from 28-30℃, whereas in Hangzhou, where this study was conducted, the summer maximum temperature can exceed 40℃. During our experiment, the highest temperature reached 39℃, demonstrating that trees provided better cooling effects on high-temperature asphalt surfaces. Tun et al. discovered that shading reduced surface temperatures by an average of 20℃, with the maximum temperature decreasing by 40℃. Their study site had higher temperatures than ours, with the highest temperature in New South Wales, Australia, reaching 48.9℃ in January 2020 [49]. Therefore, the cooling effect of trees is largely influenced by urban heatwaves.
Furthermore, different tree attributes have varying mitigating effects on the urban surface heat island. In this study, we found that trees with larger crown have better cooling effects, consistent with Wang et al.'s results [50]. This is because, as the canopy size increases, the sky view factor below the tree decreases, reducing the amount of solar radiation received. Additionally, LAI and LAD are crucial factors. Hardin and Jensen's research showed that an increase in LAI of one unit could lower surface temperatures by 1.2℃ [51], while a 1m2/m3 increase in LAD reduced surface temperatures by 4.63℃. In our study, an increase of 0.5 units in LAD lowered surface temperatures by 0.9-2.2℃, which is similar to the previous findings. However, we found that the cooling effect of trees decreased as LAD increased due to the nonlinear reduction in transmittance. When LAD reaches a certain size, trees become denser, and increasing LAD has little impact on their light transmission, resulting in a less significant increase in shading.
In addition to the inherent properties of trees, the diverse layout of trees can also have different impacts on surface temperature. Research has found that as the number of trees increases, tree crown spacing decreases and green coverage increases. The cooling effect of the trees is best when their crowns overlap. The study results are consistent with Narimani's finding that overlapping trees have a better cooling effect [52]. At the same time, as the crowns of the trees overlap, the rate at which trees cool the environment gradually decreases. This is because the crowns of other trees weaken the shading potential of adjacent trees. Additionally, building height also has an important impact on the cooling effect of trees. Previous research has shown that buildings have a certain inhibitory effect on the shading of trees [52]. This study simulated the effects of different building heights on the cooling effect of trees, and found that the impact was relatively small. This may be due to the high solar altitude angle and the east-west orientation of the experimental streets in this study, resulting in a relatively small impact of building shadows. Further research will explore the impact of building height on tree cooling for streets oriented north-south and at different dates and times.
- (2) Please provide a better explanation for why the surrounding ground is more important. In large-scale urban environments, thermal comfort has already taken the influence of the surrounding ground into account.
- At the scale of streets, particularly on asphalt pavement, temperatures can exceed 65℃ during summer, as observed in our experiments and previous research. This leads to increased vehicle rutting and strong long-wave thermal radiation, which significantly affects both vehicular energy consumption and pedestrian thermal comfort. Small-scale studies may more accurately measure the cooling effect of trees, while in large-scale urban environmental research, satellite remote sensing is often used to retrieve surface temperature for environmental cooling. However, tree cooling effects are more focused on the overall city, and there are differences in the impact of ground temperature under different scenarios at the street scale and the cooling effect of greening. Therefore, the introduction and discussion sections of the manuscript have been devoted to exploring these relevant issues.
- (3) Section 3 to Section 5? Incorrect section numbering.
- Thank you for your reminder. The text error has been corrected.

Reviewer 2 Report
The paper has scientific and social importance, as well is well written and structured. In the attached file some observations to improve the contribution and quality of the paper: 1) descrive beeter the weather types during the field data colletion, inclutind the atmospheric systems; 2) more photographys of urban canyon is welcome and, the most importante 3) contextualize the results with internacional bibliography, as presented the authors just discribe it, it's necessary explain better the physical process od cooling and heating and compare values from other experiments to frame the results.

the paper has some typographical errors that authors must pay attention, but the english is good.
Author Response
Dear reviewer,
We sincerely appreciate the editor’s responsible work and all reviewers' thoughtful comments and suggestions. Your ideas have enabled us to improve the quality of research work and make our article more scientific and clear. We have carefully reviewed the comments and have revised the manuscript accordingly. Our responses are given in a point-by-point manner below. The manuscript was modified using track Changes. Finally, we would like to take this opportunity to wish all the journal editors and reviewers a wonderful season and a happy 2023!
Reviewer 2:The paper has scientific and social importance, as well is well written and structured. In the attached file some observations to improve the contribution and quality of the paper:
- (1) derisivebeeter the weather types during the field data collation, included the atmospheric systems
- Many thanks for yoursuggestions. (line 135) The weather data and atmospheric morphology have been added to the manuscript:
During the months of July, August, and September, Hangzhou City experiences an average precipitation of 207.9mm, 246.5mm, and 105.4mm, respectively. The mean monthly temperatures during these months are 28.1℃, 29.4℃, and 25.1℃, while the mean monthly sunshine hours are 162.5h, 228.5h, and 189.7h. During this period, the city is frequently influenced by subtropical high pressure systems, resulting in a quasi-stationary wind environment under clear skies [41].
- (2) more photographys of urban canyon is welcome
- The relevant images have been added to the manuscript. (line 135)
Before modification:
After modification:
- (3) contextualize the results with internacional bibliography, as presented the authors just discribe it, it's necessary explain better the physical process od cooling and heating and compare values from other experiments to frame the results.
- Thank you for your reminder. The discussion has been separately listed as Chapter 4 of the paper and has been discussed in conjunction with previous research。This is the revised discussion section:
Currently, research on surface heat islands is conducted at various scales. Large-scale studies often use remote sensing images to infer surface temperature and analyze the urban surface heat island effect. For example, Landsat satellite images had been used to reveal that green surfaces can provide cooling effects ranging from 5-10℃ [6,43,46]. While these studies provide valuable information for decision-makers at large scales, ground temperature inversion using remote sensing imagery still lacks precision in street-valley scales. Other studies had focused on analyzing the contributions of individual trees to the environment [21], but urban greening is a collective effort that reflects the contributions of a community; obtaining accurate data that truly represents human perceptions is challenging. This study focuses on the contribution of street-level greenery to environmental temperature, taking into account human height and environmental factors. Specifically, we investigated the cooling effects of tree attributes and spatial layout on surface temperature, providing more robust evidence for urban greenery construction from the perspectives of humans and their environment.
Simulation results indicated that trees play a crucial role in cooling the urban canyon's surface. The best cooling effect occurs between 1 p.m. and 3 p.m., consistent with Duval et al.'s findings [47]. Notably, on motorway asphalt pavements, the cooling effect reaches up to 30℃, while Sharmin et al. found that trees could cool black surfaces up to 28.1℃ [48]. This difference may be due to the fact that the summer maximum temperature in their studied city ranged from 28-30℃, whereas in Hangzhou, where this study was conducted, the summer maximum temperature can exceed 40℃. During our experiment, the highest temperature reached 39℃, demonstrating that trees provided better cooling effects on high-temperature asphalt surfaces. Tun et al. discovered that shading reduced surface temperatures by an average of 20℃, with the maximum temperature decreasing by 40℃. Their study site had higher temperatures than ours, with the highest temperature in New South Wales, Australia, reaching 48.9℃ in January 2020 [49]. Therefore, the cooling effect of trees is largely influenced by urban heatwaves.
Furthermore, different tree attributes have varying mitigating effects on the urban surface heat island. In this study, we found that trees with larger crown have better cooling effects, consistent with Wang et al.'s results [50]. This is because, as the canopy size increases, the sky view factor below the tree decreases, reducing the amount of solar radiation received. Additionally, LAI and LAD are crucial factors. Hardin and Jensen's research showed that an increase in LAI of one unit could lower surface temperatures by 1.2℃ [51], while a 1m2/m3 increase in LAD reduced surface temperatures by 4.63℃. In our study, an increase of 0.5 units in LAD lowered surface temperatures by 0.9-2.2℃, which is similar to the previous findings. However, we found that the cooling effect of trees decreased as LAD increased due to the nonlinear reduction in transmittance. When LAD reaches a certain size, trees become denser, and increasing LAD has little impact on their light transmission, resulting in a less significant increase in shading.
In addition to the inherent properties of trees, the diverse layout of trees can also have different impacts on surface temperature. Research has found that as the number of trees increases, tree crown spacing decreases and green coverage increases. The cooling effect of the trees is best when their crowns overlap. The study results are consistent with Narimani's finding that overlapping trees have a better cooling effect [52]. At the same time, as the crowns of the trees overlap, the rate at which trees cool the environment gradually decreases. This is because the crowns of other trees weaken the shading potential of adjacent trees. Additionally, building height also has an important impact on the cooling effect of trees. Previous research has shown that buildings have a certain inhibitory effect on the shading of trees [52]. This study simulated the effects of different building heights on the cooling effect of trees, and found that the impact was relatively small. This may be due to the high solar altitude angle and the east-west orientation of the experimental streets in this study, resulting in a relatively small impact of building shadows. Further research will explore the impact of building height on tree cooling for streets oriented north-south and at different dates and times.
- Thank you for highlighting in the manuscript. We have made modifications to the details highlighted in the manuscript.

Author Response
Dear reviewer,
We sincerely appreciate the editor’s responsible work and all reviewers' thoughtful comments and suggestions. Your ideas have enabled us to improve the quality of research work and make our article more scientific and clear. We have carefully reviewed the comments and have revised the manuscript accordingly. Our responses are given in a point-by-point manner below. The manuscript was modified using track Changes. Finally, we would like to take this opportunity to wish all the journal editors and reviewers a wonderful season and a happy 2023!
Reviewer #3:
- (1) What is the size of theroad and the area of the study?
- Many thanks for your comments.Street size have been added in the manuscript.The Fengqi Road, with a total length of 2.86km and a street width of 40 meters, consists of paved roads that are respectively 8 meters wide and asphalt lanes that are 24 meters wide. For our experiment, we selected a main measurement area of 130 meters in length from the aforementioned street. ( line 130)
- (2) Data collection usingwhat equipments? For how many days? How many points of data? What is A B C D E represent?
- We have supplemented the experimental instruments, experimental dates, and information on various measurement points.
|
Original Text |
Belonging Line Number |
Proposed Corrections |
|
The microclimate data collected included building surface temperature of the street valley, air temperature and humidity at 1.5 meters in both the non-motorized and motor vehicle driveways, surface temperature of the non-motorized and motor vehicle driveways, black globe temperature, wind direction, and average wind speed. |
Line151 |
The experimental dates were July 23rd, 24th, August 16th, 17th, September 9th, and 10th. Three days with less cloud interference were selected as the focus of this study for analysis. To ensure the accuracy of the data, repeated tests were conducted on three sections of the north and south sides of each street valley, with measurements taken at five different times a day, at 2-hour intervals, from 9:00 am, 11:00 am, 1:00 pm, 3:00 pm and 5:00 pm (Fig. 3). The microclimate data collected included building surface temperature (point A and A') of the street valley, air temperature and humidity at 1.5 meters in both the pavement (point B and B') and motorway (point D and D'), surface temperature of the pavement (point C and C') and motorway (point E and E') , black globe temperature, wind direction, and average wind speed. The study employed TES1361C equipment to measure air temperature and humidity, TES-1333 equipment to measure solar radiation, and FLUKE F59 equipment to measure surface temperatures of both the ground and building exteriors. For each type of measurement within each section, three sampling points were selected to avoid significant data deviations. A control group with no green coverage was also measured in each valley. |
- (3) 21 m. represent theheight of the building?
- Yes, this is the building height, and the annotation has been marked in Figure 3(line 164;).
- (4) Not present in the text.
- Thank you for your reminder. Such issues have been revised in the manuscript.
- (5) There is no discussion comparing the work with other works at allWhat the other works found using the same simulation? come to the same conclusion to recommend for planning design?
- Thank you very much for your suggestion. We have listed the discussion section separately and discussed it in conjunction with previous research. The current discussion is as follows:
Currently, research on surface heat islands is conducted at various scales. Large-scale studies often use remote sensing images to infer surface temperature and analyze the urban surface heat island effect. For example, Landsat satellite images had been used to reveal that green surfaces can provide cooling effects ranging from 5-10℃ [6,43,46]. While these studies provide valuable information for decision-makers at large scales, ground temperature inversion using remote sensing imagery still lacks precision in street-valley scales. Other studies had focused on analyzing the contributions of individual trees to the environment [21], but urban greening is a collective effort that reflects the contributions of a community; obtaining accurate data that truly represents human perceptions is challenging. This study focuses on the contribution of street-level greenery to environmental temperature, taking into account human height and environmental factors. Specifically, we investigated the cooling effects of tree attributes and spatial layout on surface temperature, providing more robust evidence for urban greenery construction from the perspectives of humans and their environment.
Simulation results indicated that trees play a crucial role in cooling the urban canyon's surface. The best cooling effect occurs between 1 p.m. and 3 p.m., consistent with Duval et al.'s findings [47]. Notably, on motorway asphalt pavements, the cooling effect reaches up to 30℃, while Sharmin et al. found that trees could cool black surfaces up to 28.1℃ [48]. This difference may be due to the fact that the summer maximum temperature in their studied city ranged from 28-30℃, whereas in Hangzhou, where this study was conducted, the summer maximum temperature can exceed 40℃. During our experiment, the highest temperature reached 39℃, demonstrating that trees provided better cooling effects on high-temperature asphalt surfaces. Tun et al. discovered that shading reduced surface temperatures by an average of 20℃, with the maximum temperature decreasing by 40℃. Their study site had higher temperatures than ours, with the highest temperature in New South Wales, Australia, reaching 48.9℃ in January 2020 [49]. Therefore, the cooling effect of trees is largely influenced by urban heatwaves.
Furthermore, different tree attributes have varying mitigating effects on the urban surface heat island. In this study, we found that trees with larger crown have better cooling effects, consistent with Wang et al.'s results [50]. This is because, as the canopy size increases, the sky view factor below the tree decreases, reducing the amount of solar radiation received. Additionally, LAI and LAD are crucial factors. Hardin and Jensen's research showed that an increase in LAI of one unit could lower surface temperatures by 1.2℃ [51], while a 1m2/m3 increase in LAD reduced surface temperatures by 4.63℃. In our study, an increase of 0.5 units in LAD lowered surface temperatures by 0.9-2.2℃, which is similar to the previous findings. However, we found that the cooling effect of trees decreased as LAD increased due to the nonlinear reduction in transmittance. When LAD reaches a certain size, trees become denser, and increasing LAD has little impact on their light transmission, resulting in a less significant increase in shading.
In addition to the inherent properties of trees, the diverse layout of trees can also have different impacts on surface temperature. Research has found that as the number of trees increases, tree crown spacing decreases and green coverage increases. The cooling effect of the trees is best when their crowns overlap. The study results are consistent with Narimani's finding that overlapping trees have a better cooling effect [52]. At the same time, as the crowns of the trees overlap, the rate at which trees cool the environment gradually decreases. This is because the crowns of other trees weaken the shading potential of adjacent trees. Additionally, building height also has an important impact on the cooling effect of trees. Previous research has shown that buildings have a certain inhibitory effect on the shading of trees [52]. This study simulated the effects of different building heights on the cooling effect of trees, and found that the impact was relatively small. This may be due to the high solar altitude angle and the east-west orientation of the experimental streets in this study, resulting in a relatively small impact of building shadows. Further research will explore the impact of building height on tree cooling for streets oriented north-south and at different dates and times.
- (6) What is acceptablerange? How many%?
- The range is basically within 5 ℃, which has been supported by relevant research.
- Evaluating the impact of tree morphologies and planting densities on outdoor thermal comfort in tropical residential precincts in Singapore
- Impact of small-scale tree planting patterns on outdoor cooling and thermal comfort
- Assessing the effects of urban street trees on building cooling energy needs: The role of foliage density and planting pattern
- (7) What does this mean? The same motorized road after planting thetrees give different cooling effect?
- There are language issues with the original text here, and modifications have been made to it.
|
Original Text |
Belonging Line Number |
Proposed Corrections |
|
However, after planting trees, the cooling effect of the paved road was higher than that of the motorized road, reaching 10℃-15℃, while the cooling effect of the motorized road was between 5.3℃-6.1℃.
|
line 220 |
With the exception of the pavement on the south side, the other three roads are less affected by building shadows. In the absence of tree planting, the average surface temperature of the three roads is similar. However, after trees were planted, the cooling effect of the paved road was higher than that of the motorway, reaching 10℃-15℃, while the cooling effect on both sides of the motorway was between 5.3℃ and 6.1℃. |
- (8) How can theycontribute?
- Their contributions and the principles of their role have been supplemented in the manuscript.
|
Original Text |
Belonging Line Number |
Proposed Corrections |
|
Notably, there are fewer data points without cooling, indicating that even in areas where trees do not provide shade, they still contribute to a certain degree of cooling
|
line 245 |
It is noteworthy that there are fewer data points without cooling, indicating that even in areas where trees do not provide shade, the transpiration of trees and their shading effect on shortwave diffuse radiation from the sky and shortwave and long-wave radiation from building surfaces also have a certain degree of cooling effect. |
- (9) Temperature can bedropped for > 30C?
- In both past studies and during the actual measurements and simulations of this study, surface temperatures were found to exceed 65° or more in the motorway, while in the trees in the cooling of the concrete of the park can reach while the trees cooled and cooled, the cooling can reach 30° at fewer points directly below the trees. This section is dedicated to adding a comparison with related studies to the discussion.
- (10) Regardless of species andcrown architecture?
- In both measurements and simulations, DCand LAD were used to represent common tree properties in southeast China. In subsequent studies, we will focus on parameters such as diameter at breast height and branch height of trees at each growth stage to investigate the effect of tree properties on cooling in a more specific and comprehensive way.
- (11) from the crown diameter 7, 9 , 11 respectively?
- Hangzhou is a subtropical region, which needs sunshine in winter and shade in summer. And overhanging trees are common greenery species with obvious main trunks and obvious apical dominance. Therefore, in the greening maintenance process, in order to form a better shade effect in summer, the staff will prune it to promote the growth of its lateral branches. In the road greening in Hangzhou, which has been under construction for many years, the maximum crown width of Pendula can reach more than 10 meters. Therefore, in this study, three crown width gradients of 7, 9, and 11 meters were designed with this as a reference.
- (12) >10 m?Meaning?meaning the number of trees should be minimized? or the size of the crown? Too confusing what is this line? 50% of?
- There were some problems with the language of the manuscript, which has been revised.
|
Original Text |
Belonging Line Number |
Proposed Corrections |
|
Therefore, in the actual planning and design, trees with LAD ≥ 1.5m2/m3 can be selected and planted at equal intervals, which is beneficial to the relief of surface heat island as well as increasing pedestrian thermal comfort. Subsequently, this study aims to delve deeper into the cooling effect of trees, taking into consideration multiple factors. The tree attributes and their spatial location relationships will be analyzed to uncover the relationship between tree cooling and its various attributes, as well as its planting layout. Mathematical methods will be employed to quantify this relationship and derive a set of models for street canyons. This will provide valuable insights for future urban development, construction, and the planning and design of plant landscapes.
|
line 274 |
On the other hand, the cooling effect of trees gradually increases with the increase of LAD. When the LAD reaches 1.5m2/m3, the peak cooling effect exceeds half of the peak difference between LAD of 2.5m2/m3 and LAD of 0.5m2/m3 (indicated by the red dashed line), and the rate of change in the cooling effect of trees begins to slow down (Fig. 8(b)). Although the cooling effect also increases with the further increase of LAD, the magnitude of the increase becomes smaller. Therefore, to alleviate the urban heat island effect and improve pedestrian thermal comfort, it is recommended to select trees with a LAD value of 1.5m2/m3 or greater in planning and design. |
- (13) What does the Dc stand for?crown diameter?
- It means Crown diameter, and the abbreviation is explained in the manuscript.
- (14) should provide thesummary of each scenarios in table how many tree show much spacing how large of the tree how height of the building in each scenario
- The simulation scenarios have been further described in text as well as in pictures.
|
Original Text |
Belonging Line Number |
Proposed Corrections |
|
The first group focused on increasing the number of trees while decreasing tree spacing across scenarios 1 to 4. The second group involved modifying the tree layout while maintaining a constant number of trees across scenarios 5 to 8. The final group maintained tree positions while altering the height of buildings on either side of the street across scenarios 9 to 12.
|
page 13, line 224 |
All groups of models used trees with a crown diameter of 11 m, height of 11 m, and LAD of 1.5 m2/m3. In the first group, the cooling effect of trees was explored by changing the tree quantity to alter the greening coverage. From S1 to S4, the distance between trees decreased by 2 m in each scenario, resulting in an increase in the number of trees. The second group consisted of 12 trees with various configurations, including single standalone trees, two trees planted together, three trees planted together, and four trees planted together. This group aimed to explore the cooling effect of different combinations of trees while ensuring the total number of trees remained constant. Finally, the third group used the same tree layout and attributes as scenario 5 but varied the building height to investigate the impact of different building heights on surface temperature. |
Figure before modification:
Figure after modification:
- (15) regardless to the leaf color,leaf thickness, leaf area, leafstructure?
- We strongly agree with your point of view that leaf color, leaf thickness, leaf area, and leaf structure are all factors that affect the ground temperature of trees. However, in this study, we kept the tree crown properties unchanged and mainly adjusted the spatial layout and building height of the trees. The main focus was to investigate the cooling effect of trees under different layout scenarios. The tree attributes you mentioned are also very worthy of attention. In our subsequent research, we will focus on these influencing factors and conduct relevant studies specifically on the attributes of trees themselves.
- (16) Why only show thesl-s4 while in the 9 has 12 scenarios and show the other groups in different kind of figure?Due to the no difference between the scenarios in each group?
- This is because the trends in changes between each group are different, and if they are placed in the same graph, the readability of the image deteriorates due to too many lines. The changes in the first group are more obvious and are shown in Figure 10. However, the changes in the second and third groups are relatively small, so they are shown in Figure 11 and Figure 12, respectively. In Chapter 4, some discussions and analyses were added to explain the reasons for these changes in relation to previous research.
- (17) Instead of putting the axis17:00 in the graph, putting the number is much easier for the reader to understand
- The graphand text has been modified. The change in green coverage is positively correlated with the change in tree cooling effect, with correlation coefficients of 0.945, 0.961, 0.978, 0.965, and 0.955 at 9am, 11am, 1pm, 3pm, and 5pm, respectively. (line 306)
Figure before modification:
Figure after modification:
- (18) What planning scenariothe author suggest to do with this urban area? From those 12 scenarios?
- The conclusion here is based on Chapter 3.2.2. By simulating trees with different attributes, it is found that trees with LAD ≥ 5m2/m3 have achieved a good cooling effect. Continuing to increase LAD resulted in a decrease in the rate of cooling changes in the trees. The 12 simulation scenarios refer to the cooling effects of trees on different surface temperatures under the simulation scenario of adjusting the spatial position of different trees. The conclusion section has been reorganized:
A combined experimental and simulation study was conducted to investigate the cooling effect of trees on the street canyon surface heat island under various conditions. The following conclusions were drawn from the results:
(1) The reliability of the ENVI-met software in simulating street canyon surface temperatures was validated through comparison with measured data.
(2) The analysis of the cooling effect of trees on the north and south lanes of the street canyon at different times indicates that on the asphalt of the motor vehicle lane, the cooling effect can reach up to 30.28℃, while on the paved road, it can reach up to 24.5℃. In the daytime, the cooling effect is consistent at 9 AM. However, between 11 AM and 3 PM, the cooling effect of trees on the northern side of the road surface ranges from 10°C to 15°C, which is 2 to 4 times that of the southern side. At 5 PM, the cooling effect on the southern side reaches 10.3°C, which is twice that of the northern side.
(3) Through an analysis of tree parameters and spatial layout variations, it has been determined that maximizing the canopy diameter and selecting trees with LAD greater than 1.5 m2/m3 can aid in reducing the surface heat island effect in urban street canyons as part of urban planning and design. Additionally, Adjusting the layout while keeping the number of trees constant has a relatively minor impact on the average ground temperature of the road. However, increasing the number of trees and thus improving the green coverage of trees significantly enhances the cooling effect.

Round 2
Reviewer 1 Report
The reviewer appreciates the authors’ efforts in revising the manuscript. While authors still need to justify their methodology, including the computational setting, domain size, and boundary conditions in ENVI-met.
Author Response
Dear Reviewers and Editors,
We sincerely appreciate the editor’s responsible work and all reviewers' thoughtful comments and suggestions. We have carefully reviewed the comment and have revised the manuscript accordingly. The manuscript was modified using track changes. Finally, Thank you once again for your valuable contribution to our work.
Reviewer :
- The reviewer appreciates the authors’ efforts in revising the manuscript. While authors still need to justify their methodology, including the computational setting, domain size, and boundary conditions in ENVI-met.
- Many thanks for theAdditional explanations have been provided regarding the conditions that govern the ENVI-met simulation process. (line 178, line 187)
The revised content is as follows:
The geographic coordinates were set at 119.72°E and 30.23°N, with a time zone of UTC/GMT+08:00. The simulation model consists of a total of 100 x 80 x 30 grids, with dimensions of 2m x 2m x 2m for each grid. To enhance the accuracy of radiative heat transfer calculations, we employed the Indexed View Sphere (IVS) module in our simulation setup.
|
|
Simulation parameters conditions |
||
|
Jul.23 |
Aug.16 |
Sep.10 |
|
|
Max TEMP(℃) |
39.2 |
38.8 |
34 |
|
Min TEMP(℃) |
29 |
29.4 |
22 |
|
Max RH (%) |
77 |
69 |
43 |
|
Min RH (%) |
35 |
39 |
76 |
|
Foliage Short-wave Albedo |
0.3 |
0.3 |
0.3 |
|
Short-wave radiation absorption |
0.5 |
0.5 |
0.5 |
|
Duration (h) |
24 |
24 |
24 |
|
Start time (Local) |
07:00 |
07:00 |
07:00 |

Reviewer 2 Report
The manuscript has scientific and social merit, as well as contributing to advances in urban climatology, above all, to the mitigation aspects of urban afforestation. The authors performed with the requested adjustments by expanding, offering new decisive elements for understanding the studied processes.
The manuscript has scientific and social merit, as well as contributing to advances in urban climatology, above all, to the mitigation aspects of urban afforestation. The authors performed with the requested adjustments by expanding, offering new decisive elements for understanding the studied processes.
Author Response

(The authors gave the same response as above.)
